# Educational Interventions on Pregnancy Vaccinations during Childbirth Classes Improves Vaccine Coverages among Pregnant Women in Palermo’s Province

**DOI:** 10.3390/vaccines9121455

**Published:** 2021-12-08

**Authors:** Claudio Costantino, Walter Mazzucco, Nicole Bonaccorso, Livia Cimino, Arianna Conforto, Martina Sciortino, Gabriele Catalano, Maria Rosa D’Anna, Antonio Maiorana, Renato Venezia, Giovanni Corsello, Francesco Vitale

**Affiliations:** 1Department of Health Promotion Sciences, Maternal and Infant Care, Internal Medicine and Medical Specialties (PROMISE) “G. D’Alessandro”, University of Palermo, 90127 Palermo, Italy; walter.mazzucco@unipa.it (W.M.); nicole.bonaccorso@unipa.it (N.B.); livia.cimino@unipa.it (L.C.); arianna.conforto@unipa.it (A.C.); martina.sciortino@unipa.it (M.S.); gabriele.catalano19@gmail.com (G.C.); renato.venezia@unipa.it (R.V.); giovanni.corsello@unipa.it (G.C.); francesco.vitale@unipa.it (F.V.); 2HCU Obstetrics and Gynecology, Buccheri La Ferla—Fatebenefratelli Hospital, 90100 Palermo, Italy; danna.mr@gmail.com; 3HCU Obstetrics and Gynecology, ARNAS Ospedale Civico Di Cristina-Benfratelli Hospital, 90127 Palermo, Italy; maioran@alice.it

**Keywords:** maternal immunization, influenza vaccination, diphtheria-tetanus-pertussis vaccination, childbirth courses, vaccination counseling

## Abstract

Maternal immunization is considered the best intervention in order to prevent influenza infection of pregnant women and influenza and pertussis infection of newborns. Despite the existing recommendations, vaccination coverage rates in Italy remain very low. Starting from August 2018, maternal immunization against influenza and diphtheria-tetanus-pertussis were strongly recommended by the Italian Ministry of Health. We conducted a cross sectional study to estimate the effectiveness of an educational intervention, conducted during childbirth classes in three general hospitals in the Palermo metropolitan area, Italy, on vaccination adherence during pregnancy. To this end, a questionnaire on knowledge, attitudes, and immunization practices was structured and self-administered to a sample of pregnant women attending childbirth classes. Then, an educational intervention on maternal immunization, followed by a counseling, was conducted by a Public Health medical doctor. After 30 days following the interventions, the adherence to the recommended vaccinations (influenza and pertussis) was evaluated. At the end of the study 326 women were enrolled and 201 responded to the follow-up survey. After the intervention, among the responding pregnant women 47.8% received influenza vaccination (+44.8%), 57.7% diphtheria-tetanus-pertussis vaccination (+50.7%) and 64.2% both the recommended vaccinations (+54.8%). A significant association was found between pregnant women that received at least one vaccination during pregnancy and higher educational level (graduation degree/master’s degree), employment status (employed part/full-time) and influenza vaccination adherence during past seasons (at least one during last five years). The implementation of vaccination educational interventions, including counseling by healthcare professionals (HCPs), on maternal immunization during childbirth courses improved considerably the vaccination adherence during pregnancy.

## 1. Introduction

Pregnant women and newborns are more vulnerable to several infections, some of which are associated with elevated morbidity and mortality [1].

Vaccination of pregnant women could protect them, their fetuses and infants from several vaccine preventable diseases [2].

Guidelines on vaccination in pregnancy focus on the possible risks associated with the use of vaccines versus the benefits they might bring [1,2].

The increasing evidence of the safety of vaccines and, particularly, the absence of an increased risk of adverse events during pregnancy associated with their administration, have improved the awareness of maternal immunization across the scientific community and the general population [3].

Currently, inactivated influenza and diphtheria-tetanus-pertussis acellular (DTPa) vaccines are strongly recommended during pregnancy, as indicated by Advisory Committee on Immunization Practices (ACIP) and several International Public Health Authorities [4,5].

Influenza infections can result in serious disease/illness for mother, fetus and infant, including death [6,7]. Pregnant and postpartum women can be affected by severe forms of influenza, as reported during the “Spanish” flu in 1918/1919, Asian flu in 1956 or during AH1N1 influenza pandemic in 2009 [8,9,10,11,12]. Serious medical complications due to influenza infection during pregnancy are the result of increases in heart rate, stroke volume, oxygen consumption, and decreases in lung capacity [13,14,15,16]. Furthermore, influenza during pregnancy was significantly associated with neural tube defects, soft palate closure, posterior palate closure, and congenital cardiovascular abnormalities in infants (whether contracted in the first trimester or throughout the entire pregnancy duration) [17].

Pertussis infection among children remains a major health problem worldwide [18]. Between 2000 and 2017, the CDC reported 307 deaths due to pertussis infection in the US and 84.0% of these deaths occurred in children younger than 2 months of age [18].

Vaccine availability is not a guarantee of vaccine uptake, particularly during pregnancy. Internationally, vaccination campaigns have been differently accepted by pregnant women, depending on their perceived need, safety and efficacy, as well vaccine confidence and socio-economic index [19,20,21,22].

Vaccination counseling could improve attitudes toward vaccination during pregnancy [7,21], and it could be taken into account to avoid missed opportunities training session dedicated to vaccination as a part of childbirth classes. In Italy, vaccination counseling in pregnancy is a rather recent practice that sees its rationale in the recommendation of vaccinations in pregnancy given by the Ministry of Health in August 2018 [23,24].

Despite this recommendation, at the end of 2019 vaccination coverage against influenza and DTPa observed in the Province of Palermo—the most populous area of Sicily region and the fourth for demographic density in Italy, with 1,214,291 inhabitants and about 10,000 births in the last 10 years [25]—was of 0.2% and 2.4%, respectively (data from Sicilian Regional Health Department) [23].

The aim of this study was to evaluate the efficacy of an educational intervention, conducted during childbirth classes held in three general hospitals of the Palermo metropolitan area (Italy), to improve the vaccination adherence during pregnancy.

These three hospitals accounts for more than 40% of overall childbirths in the Province of Palermo in 2019 [26]. This study also examined the factors associated with higher vaccination acceptance rates during pregnancy.

## 2. Materials and Methods

### 2.1. Study Design

A cross sectional study with a pre/post interventional survey was carried out from October 2019 to October 2020. To this end, a self-administered anonymous questionnaire was structured and proposed to pregnant women in order to investigate their knowledges, attitudes and immunization practices during pregnancy.

After the questionnaire administration, an educational intervention focused on maternal immunization during pregnancy, immunization during life-course and vaccination recommended on the Italian Immunization Plan was conducted by healthcare professionals of the University of Palermo during the childbirth classes. Finally, 30 days after interventions, evaluate influenza or DTPa vaccination adherence of pregnant women were evaluated through a contact by text and/or WhatsApp messages or by email address. The study design was synthetized in Figure 1.

### 2.2. Data Collection

Women were recruited during childbirth courses organized at three general Hospital of the Palermo metropolitan area (“Civico Di Cristina-Benfratelli” Hospital, “Buccheri La Ferla Fatebenefratelli” Hospital and Palermo University Hospital).

The childbirth classes were face-to-face from October 2019 to the beginning of March 2020, and after, due to the COVID-19 pandemic, the courses were hold on online platforms (Skype ^®^, Microsoft Skype Division, Luxembourg, Luxembourg; Microsoft Teams ^®^, Microsoft, Washington, DC, USA; Zoom^®^, Zoom Video Communications Inc., San Jose, CA, USA), since the month of April 2020 until the end of the project.

Childbirth classes usually took place bi-monthly and had an average attendance of 15–20 couples per course. To all participants, initially, information that explained the goals of the study and the processing of personal data, according to Italian privacy laws, were provided. If the pregnant woman agreed to participate in the study, the informed consent form was collected and the questionnaire was administered.

Two methods of self-administration were provided: 1. through a paper form and 2. through an online form created on the Google^®^ Forms (Google, Menlo Park, CA, USA) platforms, which allowed completion of the questionnaire both during live childbirth classes or web-based ones held during the COVID-19 pandemic.

After filling out the questionnaire, an educational intervention was provided by trained staff made up of medical specialists and residents in Preventive Medicine and Public Health from the University of Palermo, Italy, during childbirth classes [24].

The educational intervention focused on maternal immunization during pregnancy, immunization during the lifetime of the patient and vaccination recommended on the vaccination schedule in the Italian National Immunization Plan.

In addition, a copy of the Vaccination Schedule of the Sicilian Region prepared by the Scientific Board of “VaccinarsinSicilia” (available also on vaccinarsinsicilia.org, accessed on 27 September 2021) was offered to all participants.

At the end of the educational intervention conducted during childbirth classes, which usually lasted one hour, participants had the opportunity to express any doubts or concerns about the topics covered and a further vaccination counseling “on demand” was provided if requested.

Lastly, 30 days after the educational intervention, pregnant women were contacted by text and/or WhatsApp—Meta, Inc.—Cambridge, MA, USA messages or by email (depending on whether they provided phone number or mail address) in order to evaluate influenza or DTPa vaccination adherence and main reason for vaccines refusal.

### 2.3. Questionnaire Structure

The reliability and validity of the questionnaire was evaluated in a preliminary pilot testing study conducted among 30 pregnant women. In this study, Cronbach’s alpha was calculated and corresponded to 0.79, with an adequate reliability of the test.

The questionnaire included 36 items, divided into five sections as follows:Demographic information and educational level, including gender, age, work activities, family members.Pregnancy history (proximate and remote): trimester of pregnancy, number of pregnancies to term, number of abortions, use of contraceptive methods.Self-knowledge about immunity status to Measles, Rubella, and Hepatitis B of pregnant women.Knowledge and attitudes about flu vaccination and DTPa vaccination during pregnancy. This section was further divided in two other subsections:
–*Knowledge and attitudes to influenza and flu vaccination during pregnancy*: duration of flu season in Sicily, knowledge about flu vaccine and recommendation for pregnant women, suggested period for vaccination during pregnancy, potential complication of influenza infection during pregnancy, willingness to vaccinate themselves against flu (Likert scale from 0 to 10).–*Knowledge and attitudes on DTPa vaccination during pregnancy*: knowledge about pertussis vaccine and recommendation for vaccination among pregnant women, potential complication of pertussis infection for the newborn, willingness to vaccinate themselves with *DTPa* (Likert scale from 0 to 10).Knowledge and attitudes about early childhood vaccination.

### 2.4. Statistical Analysis

Absolute and relative frequencies were calculated for the categorical (qualitative) variables. The differences in the categorical variables for hesitancy or refusal and between before and after the intervention were analyzed using chi-squared tests (or Fisher test when required).

All the variables found to have a statistically significant association with vaccination during pregnancy refusal/acceptancy in the univariate analysis were included in a multivariate backward stepwise logistic regression model.

In addition, all the variables with a *p*-value ≤0.20 were selected in the multivariate model, to guarantee a more conservative approach.

Crude odds ratios (ORs) and adjusted ORs (adj-ORs) with their 95% confidence intervals (CIs) were calculated. The level of significance chosen was 0.05 (two tailed).

A database was created with EpiInfo 3.5.4 (Centers for Disease Control and Prevention, Atlanta, GA, USA) and all the data were analyzed using the statistical software package Stata/MP 14.1 (StataCorp LP, College Station, TX, USA).

### 2.5. Ethical Considerations

Personal data (such as the phone number or the e-mail address necessary for the contact after 30 days from the course) were collected for research purposes only and all data obtained were analyzed in aggregate form and processed in accordance with Articles 13–14 of EU Regulation GDPR 2016/679 for the protection of individuals with regard to the processing of personal data. The study was approved by the Palermo Ethical Committee 1 of the University Hospital of Palermo in session no. 9 in October 2019.

## 3. Results

Three hundred and twenty-six (*n* = 326) pregnant women were enrolled and completed the proposed questionnaire. The main sociodemographic and pregnancy data are represented in Table 1. The age group most represented was the 25–34 year old (66%), followed by the 35–40 year old age group (28.8%). The under-25 age bracket was underrepresented (1.5%) as was the percentage of women over 40 (3.7%).

With regard to educational level, 60.7% of women had a degree equal to or greater than a university degree, followed by 33.1% of participants with a high school license and 6.1% with primary or secondary school license. Most of the women interviewed were employed (full-time 45.4%, part-time 22.4%), while only about a third of them were not employed (22.7%) or were housewives (9.5%).

The majority of subjects enrolled were in the third trimester of gestation (90.2%) and at the first pregnancy (89%). Usually, in Sicily, childbirth classes involved pregnant women at the end of the second or at the third trimester of pregnancy.

Furthermore, 80.4% of the interviewed women stated that they had not had an abortion or a previous at-term pregnancy.

The analysis of the immunization/vaccination status against the hepatitis B showed that less than 60% of women were immunized against, 38.3% did not know their immunization status and 5.2% were not immunized (Table 2).

With regard to measles and rubella, 42% were naturally immunized, with contracted the diseases, 7.1% and 3.1% were not immunized and 7.7% and 15.6% did not known their immunization status against rubella and measles, respectively (Table 2).

In Table 3, knowledge, attitudes and practices of pregnant women with regard to influenza and DTPa vaccinations were reported. Only 20.8% of the participating women were aware of the beginning of flu season in Sicily. More than half (57.1%) were aware of the possible impact of influenza severe complications on mother, fetus, and newborn. Around 70% of respondents were informed about the recommendation of flu vaccination during pregnancy, while only 23.9% of the sample was aware that influenza vaccination in pregnancy could be administrated during all trimesters of gestation.

Investigating the previous adherence to influenza vaccination, the majority of respondents resulted in be never vaccinated and only 23.3% were vaccinated 1 or 2 times during the last five influenza seasons.

Regarding the possible complications resulting from the infection with *B. Pertussis*, only 36.5% of the respondents knew the possible impact of the diseases among newborns. More than half (58.6%) were aware about the recommendation of DTPa during pregnancy while 54.6% of the sample did not know the correct period for being vaccinated during pregnancy. Only, 32.8% of the pregnant women were aware that a booster of the DTPa vaccine should be repeated at any pregnancy.

Around one half (47.9%) of the responding women did not receive or search any information on vaccination in pregnancy. On the other hand, 27.9% reported that health care professionals (general practitioner, obstetrician/gynecologist, pediatricians, etc.) represented the main information sources, followed by official (Ministry of Health, etc.) websites (24.2%).

Lastly, only 10 (3.1%) and 24 (7.4%) pregnant women were vaccinated against influenza and DTPa, respectively, before the administration of the questionnaire and the vaccination counseling intervention.

Overall, 201 (62%) respondents provided a feedback to the follow-up brief survey after 30 days from the educational intervention, administered through the contact information they provided (mail or text message). The remaining 125 pregnant women (38%) that participated to the childbirth classes and received the vaccination counseling did not provide information regarding whether or not they had received flu and anti- DTPa vaccinations (data not shown).

Among the 201 responding pregnant women 96 (47.8%; +44.8%) stated that during current pregnancy, and after the intervention, they had undergone influenza vaccination, 116 (57.7%; +50.7%) declared to have received DTPa vaccination, and 129 (64.2%; +54.8%) stated that they had undergone at least to one of vaccinations recommended (Figure 2).

The main reasons reported for vaccine refusal among women interviewed, 30 days after the educational intervention, were: fear of adverse events (47.6%), vaccinations were not recommended by gynecologist/obstetrician (43.4%) and, only for influenza vaccination, the intervention was conducted outside the seasonal influenza vaccination campaign (9%) (data not shown in figure).

In Table 4 are reported the results of multivariate analysis of factors associated with influenza or DTPa vaccinations adherence (vs. vaccination refusal) among 201 pregnant women that responded to the follow-up survey.

A significant statistical association was found between adherence to at least one recommended vaccination and higher educational level (Adj OR = 3.12; 95%CI: 1.25–4.67), type of employment (part/full-time) (Adj OR = 1.89; 95% CI: 1.11–5.23), with good knowledge and information about influenza infection and vaccination (Adj OR = 1.69; 95% CI: 1.14–2.21), with good knowledge and information about pertussis infection and DTPa vaccination (Adj OR = 1.48; Confidence Interval 95% 1.08–2.12), receiving at least one vaccination during last five influenza seasons (Adj OR = 4.12; Confidence Interval 95% 2.06–5.46).

## 4. Discussion

The main of the present study was to evaluate the efficacy of an educational intervention conducted during childbirth to improve the vaccination adherence during pregnancy. At the same time, we aimed at identifying factors that may or not lead to vaccination adherence in order to propose a targeted educational intervention, to improve awareness of the importance of maternal immunization.

Maternal immunization is an effective act of health promotion both for pregnant women and newborns. Although the benefits of maternal immunization against flu and whooping cough are well-known and supported by evidence, its implementation has not yet entered into routine practice and it is often limited due to cultural and organizational barriers [27].

A recent study by the University of Colorado detecting reasons for rejection of recommended vaccinations in pregnancy, analyzed responses of 331 between obstetricians and gynecologists [28]. Overall, an adherence for DTPa vaccination greater than 10% in comparison with influenza vaccination was reported among pregnant women followed by HCPs involved in the study, similarly to what we observed in our setting.

More in depth, obstetricians and gynecologists reported as the main reason for higher DTPa vaccine uptake of their patients, the better knowledge of pregnant women on this vaccination. In general, a lower vaccination coverage rate against influenza could be attributable to the limited time range of flu vaccination offer during the year (from October to January).

In the present study, among pregnant women that responded to the follow-up survey, 47.8% carried out influenza vaccination and 57.7% DTPa vaccination (64.2% of interviewed received at least one of the two vaccination recommended). These results show a considerable improvement in vaccination adherence in comparison with data reported in Sicily by the Regional Health department due to the educational interventions conducted (<2.5% for DTPa and lower for influenza vaccination).

It’s arguable that the introduction of vaccination counseling interventions on maternal immunization during childbirth courses in the three main Hospitals of the Province of Palermo involved in the study could have positively affected vaccination coverage.

In United Kingdom, during the 2018/2019 season, vaccine coverage in pregnant women against pertussis was higher than 70% and against flu was 45.2%, while in the US 61.2% of pregnant women received influenza vaccination during 2019/2020 season, rate similar to what observed in our sample after the educational interventions conducted [29].

A recent multicenter study investigated the acceptability of antenatal vaccination among patients and the HCPs level of confidence in vaccine recommendations.

This study found that according to HCPs, the most commonly cited reasons for declining maternal immunization were concerns regarding possible side effects for the baby, doubts regarding efficacy and lack of knowledge about the impact of influenza disease [30].

Additionally in the US study the most commonly reasons for vaccine refusal, reported by obstetricians or gynecologists, were patients’ doubt on safety and efficacy of maternal immunization, similarly to what observed in our context [28].

Around 30% and 40% of respondents did not know the recommendation from the Italian Ministry of Health for flu and pertussis vaccinations in pregnancy, respectively. More than half of the study participants did not know the recommended period for DTPa vaccination, and only 24% indicated the correct period to receive influenza vaccination during pregnancy. Generally, the lack of reliable information is one of the main causes of poor adherence to recommended vaccinations in pregnancy.

As with that reported in previous studies, healthcare professionals (HCPs) recommending or not recommending vaccinations could play a decisive role in patients’ choice [31]. In the present analysis, 43% of pregnant women that refused vaccination after the formative intervention were discouraged by their obstetrician or gynecologist.

In this context, beyond of the educational contribution of HCPs [32], in the light of the increasing need for task shifting both in high and low income settings, the supportive role of the obstetrician or gynecologist should be taken into account as well [33].

The significant association between adherence to recommended vaccination in pregnancy and higher educational level and having employment (part/full-time) could confirm the role of socio-economic and educational level on correct health choices [34].

As with what reported is elsewhere, having a low-middle education level among pregnant women was associated with lower vaccine uptake during pregnancy, in comparison to women with high education level.

Usually, women with a higher education level demonstrated a greater propensity to find reliable medical information and to trust in vaccination [34,35].

Moreover, in our sample, a significant association with higher adherence to vaccination in pregnancy was found with women who receipt of at least one flu vaccination during last five influenza seasons.

A recent study conducted in Spain confirmed that women who usually adhere to influenza vaccination campaign are more prone to adhere to influenza vaccination during pregnancy [33,36].

During pregnancy, the combination of correct information and consistent recommendations provided by healthcare professionals can be considered to be one of the strongest predictive factors, increasing the chances of adherence to highly recommended vaccinations during pregnancy and the education of gynecologists and obstetricians on vaccination themes could improve in future vaccines adherence during pregnancy [37,38].

There are some limitations of the study that need to be highlighted.

First, a possible lack or representativeness due to the limited number of participants and that the enrollment process has not been in the control of researchers should be considered. Nevertheless, all pregnant women that attended childbirth classes participated in the study (>99%) and the hospitals involved in our research accounted for more than 40% of birth in the Palermo’s Province.

Another possible limitation could be attributable to a selection bias, whereas pregnant women that participate to childbirth classes have a higher educational level in comparison with the general population (only 6% with Primary/Secondary school graduation in our sample vs. 34% of 18–45 year old Sicilian women).

During the follow-up study, the percentage of women with Primary/Secondary school graduation did not change significantly (8%). Moreover, in our sample, vaccination adherence before the educational intervention conducted was better than observed in general population (3% for influenza and 7% for DTPa vaccination), but considerably lower than recommended by Italian Ministry of Health.

Therefore, the implementation of interventions and counseling programs should be broadly promoted, despite socio-economic status or educational level.

## 5. Conclusions

Educational interventions on vaccination represent a well-known preventive tool that could be very useful in the promotion of strategies limiting infections in pregnant women and newborns.

Unfortunately, in recent years, despite the strong recommendation provided by health authorities, Italy has failed to bring vaccination coverage among pregnant women to the levels indicated and recommended, contributing to avoidable morbidity and mortality, as well as an increase in health care costs.

Lack of correct information and knowledge about maternal immunization are the main reason for vaccine hesitancy within pregnant women, and could be also attributable to a lack of counseling from healthcare professionals.

Moreover, a large proportion of pregnant women refused vaccination because their obstetrician and gynecologist discouraged them. This is a serious issue and intervention to educate these doctors is likely to be as important as participant education.

Therefore, the permanent inclusion of educational interventions by teams of trained healthcare professionals, during pre-partum courses, on the topic of maternal immunization, but also on the importance of the cocoon strategy and neonatal immunization, should be standardized and extended to the regional and national level in order to increase vaccination adherence during pregnancy and during post-partum. At the same time, educating and training gynecologists and obstetricians on vaccination themes could further improve in future vaccines adherence during pregnancy.

## Figures and Tables

**Figure 1 vaccines-09-01455-f001:**
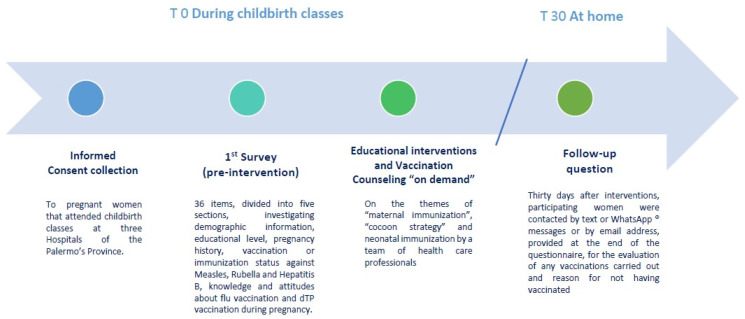
Timeline of the study design at T0 (during childbirth classes) and at T30 (pregnant women participating from home).

**Figure 2 vaccines-09-01455-f002:**
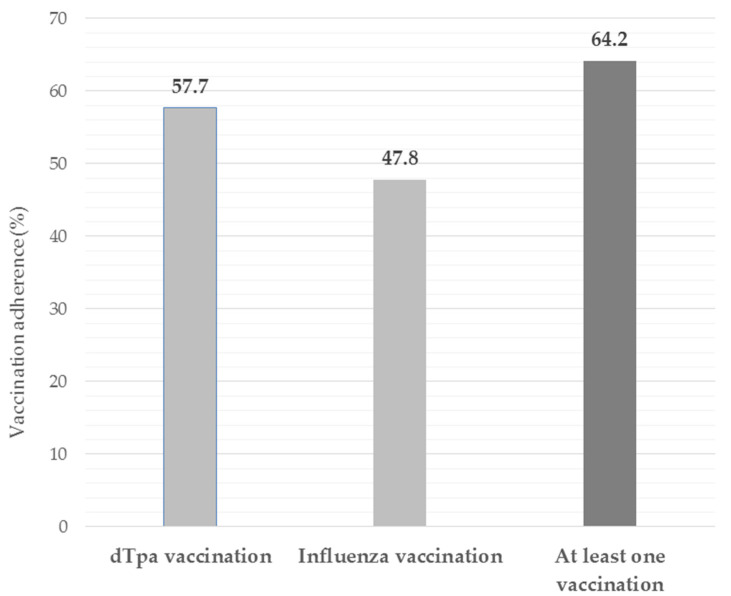
Adherence (%) to recommended vaccinations during current pregnancy following the counseling intervention, among the 201 pregnant women responding to the follow-up questions.

**Table 1 vaccines-09-01455-t001:** Socio-demographic characteristics and data related to pregnancy of the 326 pregnant women recruited in the study.

Age in Years	*n* (%)
18–24	5 (1.5)
25–34	215 (66)
35–40	94 (28.8)
≥40	12 (3.7)
**Level of education**	*n* (%)
Primary/Secondary school license	20 (6.1)
High school license	108 (33.1)
Graduation/Post graduate	198 (60.7)
**Employment**	*n* (%)
Housewife	31 (9.5)
Unemployed	74 (22.7)
Full-time employed	148 (45.4)
Part-time employed	73 (22.4)
**Pregnancy trimester**	*n* (%)
I	4 (1.2)
II	28 (8.6)
III	294 (90.2)
**Parity**	*n* (%)
0	290 (89)
1	33 (10.1)
2	1 (0.3)
≥3	2 (0.6)
**Abortions or previous at-term pregnancies**	*n* (%)
0	262 (80.4)
1	52 (16)
2	9 (2.8)
≥3	1 (0.3)

**Table 2 vaccines-09-01455-t002:** Immunization status of the 326 pregnant women in study against the main virus vaccine preventable diseases (VPDs) at the time of questionnaire completion.

	Vaccinated in Pediatric/Adolescent Age *n* (%)	Naturally Immunized With Contracted the Disease *n* (%)	Vaccinated in Adulthood in Forecast of Pregnancy or Vaccinated after Previous Unprotected Pregnancy *n* (%)	Not Immunized *n* (%)	Unkwown *n* (%)
HBV	166 (50.9)	3 (0.9)	15 (4.6)	17 (5.2)	125 (38.3)
Measles	115 (35.3)	137 (42)	13 (4)	10 (3.1)	51 (15.6)
Rubella	109 (33.4)	137 (42)	32 (9.8)	23 (7.1)	25 (7.7)

**Table 3 vaccines-09-01455-t003:** Knowledge, attitudes, and practices about influenza and DTPa vaccinations during pregnancy of the 326 women in study.

**Start of Influenza Season in Sicily**	***n* (%)**
Correct answer (usually December)	68 (20.8)
Incorrect answer	258 (79.2)
**Impact of influenza infection during pregnancy**	*n* (%)
Correct answer (a serious conditions for mother, fetus and newborn in the first months of life)	186 (57.1)
Incorrect answer	140 (42.9)
**Start of influenza vaccination campaign in Sicily**	*n* (%)
Correct answer (usually October–November)	165 (50.6)
Incorrect answer	161 (49.4)
**Recommendation of influenza vaccination during pregnancy**	*n* (%)
Correct answer (Yes)	228 (69.9)
Incorrect answer (No)	98 (30.1)
**Trimesters of pregnancy during which flu vaccination was recommended**	*n* (%)
Correct answer (all trimesters)	78 (23.9)
Incorrect answer	248 (76.1)
**Vaccinated against influenza during past five flu seasons**	*n* (%)
Never	250 (76.7)
Rarely (1 or 2 times)	76 (23.3)
**Impact of pertussis infection during newborn period**	*n* (%)
Correct answer (a serious conditions newborn during first months of life)	119 (36.5)
Incorrect answer	207 (63.5)
**Recommendation of DTPa vaccination during pregnancy**	*n* (%)
Correct answer (Yes)	191 (58.6)
Incorrect answer (No)	135 (41.4)
**Period of pregnancy during which DTPa vaccination was recommended**	*n* (%)
Correct answer (from the 27th to the 36th week of gestation)	148 (45.4)
Incorrect answer	178 (54.6)
**DTPa vaccination booster during pregnancy**	*n* (%)
Correct answer (during each pregnancy)	107 (32.8)
Incorrect answer	219 (67.2)
**Main source of information on recommended vaccination in pregnancy**	*n* (%)
Healthcare professionals (general practitioner, obstetrician/gynecologist, pediatricians)	91 (27.9)
Official website (Ministry of Health, VaccinarSi website, etc.)	79 (24.2)
No information sources	156 (47.9)
**Influenza vaccination receipt before the childbirth classes**	*n* (%)
Yes	10 (3.1)
No	316 (96.9)
**DTPa vaccination receipt before the childbirth classes**	*n* (%)
Yes	24 (7.4)
No	302 (92.6)

**Table 4 vaccines-09-01455-t004:** Factors associated with influenza and DTPa vaccination adherence (receiving at least one of the recommended vaccination during pregnancy in the month following the educational interventions) among pregnant women responding to the follow-up survey (129 women vaccinated with at least one vaccine on 201 that responded).

Vaccination Adherence (Receiving at Least One of the Recommended Vaccination) after Educational Interventions				
	Crude OR (95% CIs)	*p*-Value	Adj OR (95% CIs)	*p*-Value
**Age**				
- ≤ 35 years	ref			
- >35 years	1.26 (0.68–2.33)	0.46		
**Education level**				
- high school/ primary-secondary school degree	ref	<0.001	ref	<0.01
- graduation degree/master degree	2.89 (1.52–4.96)		3.12 (1.25–4.67)	
**Employment**				
- housewife/unemployed	ref	<0.001	ref	<0.05
- employed part/full-time	1.25 (1.49–5.05)		1.89 (1.11–5.23)	
**Number of previous pregnancies/abortions**				
- 0	ref	0.36		
-≥1	1.55 (0.58–1.63)			
**Main source of information on recommended vaccination in pregnancy**				
- No information sources	ref	0.11	ref	0.15
- Official websites	1.22 (0.81–1.69)		1.18 (0.67–1.59)	
- Healthcare professionals	1.56 (0.95–2.12)		1.49 (0.89–1.86)	
**Correct information about influenza infection and flu vaccination during pregnancy**				
- No (≤ 3 correct answers of 5 questions)	ref	<0.01	ref	<0.01
- Yes (≥ 4 correct answer of 5 questions)	1.78 (1.15–2.69)		1.69 (1.14–2.21)	
**Correct information about pertussis infection and dTpa vaccination during pregnancy**				
- No (≤ 2 correct answers of 4 questions)	ref	<0.05	ref	<0.05
- Yes (≥ 3 correct answers of 4 questions)	1.56 (1.08–2.12)		1.48 (1.08–2.12)	
**Influenza vaccination (at least one during last five years)**				
- No	ref	<0.001	ref	<0.001
- Yes	4.25 (2.27–5.75)		4.12 (2.06–5.46)	

## Data Availability

Data available on request due to restrictions of privacy.

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
