# Peer review of "Educational Interventions on Pregnancy Vaccinations during Childbirth Classes Improves Vaccine Coverages among Pregnant Women in Palermo’s Province"

_vaccines, 2021, doi:10.3390/vaccines9121455_

Round 1

Reviewer 1 Report

Line 169: It would be good to have the number also in parenthesis

line 213:  Correction to the word pregnancy

line 214: one half instead of an half

line 236: In table 4 is only the multivariate analysis

line 238 : It would be useful to present a table with the main reasons of vaccine refusal 

In line 298 you are mentioning that "the healthcare professionals recommending or not vaccinations could play a decisive role in patients’ choice ". Would it be possible to compare  between the pregnant women who were vaccinated,  the percentages of those who were instructed by their healthcare professionals and those who were not, to see if there is  statistical significance? 

Author Response

Reviewer #1

A: Dear reviewer,

Thank you for the opportunity to revise our manuscript and for appreciating the original article “Recommending Vaccination in Pregnancy: The Role of Dedicated Educational and Counselling Interventions during Childbirth Classes in Palermo’s Province” submitted to Vaccines.

Your useful comments were considered with attention and a point by point answer to your remarks and questions was reported below.

Comment: Line 169: It would be good to have the number also in parenthesis

A: The correction was carried out.

Comment: Line 213:  Correction to the word pregnancy

A: Thank you, the word was corrected

Comment: line 214: one half instead of an half

A: Thank you for your suggestion, the term was corrected.

Comment: line 236: In table 4 is only the multivariate analysis

Finally, vaccination coverage rates were calculated as usual.

Comment: line 238: It would be useful to present a table with the main reasons of vaccine refusal

A: Thank you for your suggestion, in the results section we add the present sentence: “The main reasons reported for vaccine refusal among women interviewed 30 days after the educational intervention were: fear of adverse events (47.6%), vaccinations were not recommended by gynecologist/obstetrician (43.4%) and, only for influenza vaccination, the intervention was conducted outside the seasonal influenza vaccination campaign (9%).” 

Comment: In line 298 you are mentioning that "the healthcare professionals recommending or not vaccinations could play a decisive role in patients’ choice". Would it be possible to compare between the pregnant women who were vaccinated, the percentages of those who were instructed by their healthcare professionals and those who were not, to see if there is  statistical significance? 

A: Thank you for your suggestions. Unfortunately, among vaccinated women we did not ask the main reason for vaccination adherence. We could argue that a decisive role could be attributed to the formative interventions conducted. Moreover, the data obtained from pregnant women that refused vaccination (added in results section) support another consideration, in line with your very useful suggestion, that we add in discussion section: “In the present study, 43% of pregnant women that refused vaccination after the formative intervention were discouraged by their obstetrician or gynecologist. At the same time, the dedicated formative and counselling interventions conducted during childbirth classes could support the importance of HCPs recommendations.”

Reviewer 2 Report

The first question comes for lines 30-33. This is a trial study amd you care compare means before and after the intervention. Why association has been reported that belongs to cross-sectional studies? It can be described in the methods section as the reason for choosing this style of data analysis.

Throughout the text, you need to be precise with who 'specialized personel' are.  Start with abstract and go on through the text.

The flow of the information provided in the methods section will be better if you organise this section and provide related information under subheadings as follows: design, sample and setting, data collection, data analysis, ethical considerations. Please be elaborative in the provision of details in such a manner that other authors can repeat your study.

A flow diagran can help with depicting the research method.

Please describe in more details the educational process taken place in the online courses. Describe Who has done what and how etc.

The process of development of the instruments for data collection should be described.

 The research limitations should be fully described as the end of discussion.

Author Response

Reviewer #2

Dear reviewer,

Thank you for the opportunity to revise our manuscript. We hope that this revised version could improve the article and may have solved the major problems raised by your revision.

Your useful comments were considered with attention and a point by point answer to your remarks and questions was reported below.

Comment: The first question comes for lines 30-33. This is a trial study amd you care compare means before and after the intervention. Why association has been reported that belongs to cross-sectional studies? It can be described in the methods section as the reason for choosing this style of data analysis.

A: Thank you for your observation. The type of the study was corrected in abstract and methods. As you can see in the text, the post interventional survey consist of only question related to vaccine administration, and if not, the main reason for refusal conducted via WA or text messages (and not of the entire questionnaire administered before the intervention). Mainly for this reason a comparison of means before and after intervention was not possible.

Comment: Throughout the text, you need to be precise with who 'specialized personel' are.  Start with abstract and go on through the text

A: Thank you for your suggestion. We have specified the specialized personel that conducted the interventions. In the abstract

Comment: The flow of the information provided in the methods section will be better if you organise this section and provide related information under subheadings as follows: design, sample and setting, data collection, data analysis, ethical considerations.

Please be elaborative in the provision of details in such a manner that other authors can repeat your study.

A: In accordance with your useful suggestions, we organize methods section in subheading, and we also better elaborate this section in order that other authors can repeat the present study with a flow diagram of the research method.

Comment: A flow diagran can help with depicting the research method.

A: As previously stated, we add a flow diagram in the methods section in order to better explain the study details.

Comment: Please describe in more details the educational process taken place in the online courses. Describe Who has done what and how etc.

A: As correctly suggested, the study design was better specified in the methods section and also in the new chart that better explain the timeline of the interventions.

Comment: The process of development of the instruments for data collection should be described.

A: Following your useful suggestion, in the subsection of methods regarding the questionnaire we better explain the pilot study and data obtained.

Comment: The research limitations should be fully described as the end of discussion.

A: As correctly suggested we described all the main study limitations at the end of the discussion section

Round 2

Reviewer 2 Report

Your article has been improved, but some more changes are needed:

Lines 192-238, the whole sentences do not make sense and needs editing. Also, there are two 'after the intervention' here.

Given that you have not planned for the intervention, you should clearly describe it in the methods section that you have done this cross-sectional study on the running educational intervention at your healthcare setting. 

In the abstract and also in the results, you have stated 'interviews' have been conducted, but nothing about it has been stated in the methods section.

Lines 253-257, please specify how the data practically has been collected. Did you asked the questions and wrote answers? OR the participants filled out the questionnaire themselves? You need to be very practical with the data collection process.

Go through the Equator website and fill out the STROBE checklist and ensure that all details for reporting cross-sectional studies. The Strengthening the Reporting of Observational Studies in Epidemiology (STROBE) Statement: guidelines for reporting observational studies | The EQUATOR Network (equator-network.org)

Please add a copy of the data collection tool as the supplementary file.

Add to the limitations, that the intervention process has not been in the control of researchers. Therefore, you can not be sure of the representativness of the samples etc. Suggestions for future research should be stated. 

Author Response

Answer to Reviewer #2

Comment: Your article has been improved, but some more changes are needed:

A: Dear reviewer,

Thank you for appreciating our first round of review and for the suggestions that could contribute to improve further our manuscript.

Your useful comments were considered with attention and a point by point answer to your remarks and questions was reported below.

Comment: Lines 192-238, the whole sentences do not make sense and needs editing. Also, there are two 'after the intervention' here.

A: Thank you for the suggestion, the whole sentences were edited and improved in their contents.

Comment: Given that you have not planned for the intervention, you should clearly describe it in the methods section that you have done this cross-sectional study on the running educational intervention at your healthcare setting.

A: We agree with your comment and we specified it in the methods section.

Comment: In the abstract and also in the results, you have stated 'interviews' have been conducted, but nothing about it has been stated in the methods section.

A: Thank you for your suggestion, the correction are carried out.  

Comment: Lines 253-257, please specify how the data practically has been collected. Did you asked the questions and wrote answers? OR the participants filled out the questionnaire themselves? You need to be very practical with the data collection process.

A: Thank you for your suggestion. We specified better than the questionnaire were self-administered both in presence (until March 2020) than through Google® Forms platforms for web-based classes (starting from April 2020)

Comment: Go through the Equator website and fill out the STROBE checklist and ensure that all details for reporting cross-sectional studies. The Strengthening the Reporting of Observational Studies in Epidemiology (STROBE) Statement: guidelines for reporting observational studies | The EQUATOR Network (equator-network.org)

A: Thank you for your useful suggestion. We revised the STROBE checklist and we add in abstract, introduction, methods, results and discussion section all the missing and requested details.

Comment: Please add a copy of the data collection tool as the supplementary file.

A: Thank you for your request. The questionnaire was submitted as supplementary file.

Comment: Add to the limitations, that the intervention process has not been in the control of researchers. Therefore, you can not be sure of the representativness of the samples etc. Suggestions for future research should be stated. 

A: Thank you for your useful suggestion. The limitation was added and suggestion for future research was also added in the discussion section